# Segmentation of Pulp and Pulp Stones with Automatic Deep Learning in Panoramic Radiographs: An Artificial Intelligence Study

**DOI:** 10.3390/dj13060274

**Published:** 2025-06-19

**Authors:** Mujgan Firincioglulari, Mehmet Boztuna, Omid Mirzaei, Tolgay Karanfiller, Nurullah Akkaya, Kaan Orhan

**Affiliations:** 1Department of Dentomaxillofacial Radiology, Faculty of Dentistry, Cyprus International University, Nicosia 99010, Cyprus; mujganfirincioglulari@gmail.com; 2Department of Dentomaxillofacial Surgery, Faculty of Dentistry, Cyprus International University, Nicosia 99010, Cyprus; mehmet_boztuna@yahoo.co.uk; 3Department of Biomedical Engineering, Faculty of Engineering, Near East University, Mersin 10, Lefkoşa 99010, Turkey; omid.mirzaei@neu.edu.tr; 4Department of Management Information Systems, School of Applied Sciences, Cyprus International University, Nicosia 99010, Cyprus; tkaranfiller@ciu.edu.tr; 5Dentmetria A.Ş., İstanbul 34726, Turkey; nurullah.akkaya@dentmetria.com; 6Department of Dentomaxillofacial Radiology, Faculty of Dentistry, Ankara University, Ankara 06100, Turkey

**Keywords:** artificial intelligence, deep learning, panoramic radiograph, pulp stone

## Abstract

**Background/Objectives**: Different sized calcified masses called pulp stones are often detected in dental pulp and can impact dental procedures. The current research was conducted with the aim of measuring the ability of artificial intelligence algorithms to accurately diagnose pulp and pulp stone calcifications on panoramic radiographs. **Methods**: We used 713 panoramic radiographs, on which a minimum of one pulp stone was detected, identified retrospectively, and included in the study—4675 pulp stones and 5085 pulps were marked on these radiographs using CVAT v1.7.0 labeling software. **Results**: In the test dataset, the AI model segmented 462 panoramic radiographs for pulp stone and 220 panoramic radiographs for pulp. The dice coefficient and Intersection over Union (IoU) recorded for the Pulp Segmentation model were 0.84 and 0.758, respectively. Precision and recall were computed to be 0.858 and 0.827, respectively. The Pulp Stone Segmentation model achieved a dice coefficient of 0.759 and an IoU of 0.686, with precision and recall of 0.792 and 0.773, respectively. **Conclusions**: Pulp and pulp stones can successfully be identified using artificial intelligence algorithms. This study provides evidence that artificial intelligence software using deep learning algorithms can be valuable adjunct tools in aiding clinicians in radiographic diagnosis. Further research in which larger datasets are examined are needed to enhance the capability of artificial intelligence models to make accurate diagnoses.

## 1. Introduction

Throughout life, pulp undergoes reactive changes to extrinsic and intrinsic factors in the mouth. Pulp stones are areas of calcified masses of different sizes that can occur in various locations within dental pulp. The literature suggests their prevalence to be between 8% and 95% [1,2,3,4]. As pulp stones do not routinely manifest with clinical symptoms or result in pulpal pathology, it is uncertain whether these calcifications should be classified as pulpal disease or biological variation [2,5]. Although the etiology of pulp stones has yet to be completely determined, researchers have identified a number of potential contributing factors, such as various systemic diseases, age, gender, chronic inflammation due to dental restorations, and orthodontic treatments [6,7,8,9,10]. While dental treatments are not indicated solely for pulp stones, their presence has been associated with complicated dental procedures and prolonged treatment duration [11]. Pulp stones can be detected with various different sizes, varying from minute particles that can only be detected microscopically to large masses that cover the entirety of the pulp [12]. While most pulp stones are round or oval with smooth contours, they can also be shaped irregularly with unusual contours [13]. Pulp stones can be observed independently within the pulpal tissue, connected to, or integrated within the dentine layer [12]. Although pulp stones can exist anywhere within the pulp, they are more commonly encountered in the coronal part of the complex [14]. Histological or radiological methods may be used to detect pulp stones [15,16,17]. Radiographs are the primary method used to diagnose pulp stones in dental clinics. Evidence shows that for pulp stones to be detected on radiographs, the size of the mass should be over >200 µm [18,19]. Misdiagnosis of pulp stones can lead to unexpected difficulties during dental treatment.

With the recent advancements in artificial intelligence (AI), its application in medicine and dentistry has increased considerably. AI benefits various dentistry areas, with the aim of enhancing decision-making in clinical environments, diagnosis, and treatment planning [20,21,22,23,24]. Deep learning, a subset of AI, utilizes algorithms that train from large data groups rather than pre-arranged instructions [25]. Convolutional neural networks (CNN) are deep learning tools that are considered to be adequate in object recognition. AI can help to minimize the misdiagnosis of pulp stones.

Identifying pulp stones on panoramic radiographs poses challenges due to intricate image structures and overlapping anatomical features, underscoring the necessity for advanced diagnostic methods. Artificial intelligence, particularly through deep learning techniques like convolutional neural networks, has demonstrated significant potential in medical image analysis, improving the accuracy, reliability, and efficiency. AI has been successfully utilized in dentistry to detect issues such as dental caries, bone loss, and periapical pathologies. However, research on using AI to identify pulp stones—especially in panoramic radiographic images—remains scarce. This underscores a vital gap in existing knowledge and reinforces the need for additional studies to create and validate AI-based diagnostic tools to help clinicians navigate treatment complexities and reduce potential complications.

AI has been applied to improve the analysis of dental radiographic images. In the context of two-dimensional (2D) radiographs, these digital images consist of numerous pixels, each with different levels of brightness and radiopacity. AI systems are trained to interpret these images by learning from these pixel-based features [26,27]. Recent advances in AI have shown promise in medical imaging; a deep learning algorithm capable of automatically detecting teeth in panoramic radiographs is regarded as a significant advancement in dental practice [28,29]. Also, lesion diagnosis using AI and automatic description of anatomic landmarks showed great results in terms of AI and panoramic radiographs [26,30,31].

We hypothesize that artificial intelligence can precisely identify pulp stones in routine panoramic radiographs, achieving a diagnostic performance similar to that of expert human interpretation, despite the inherent limitations of panoramic imaging.

According to the authors’ review of the literature, the ability of artificial intelligence to accurately diagnose pulp and pulp stones has not been studied on panoramic radiographs using the U2-Net architecture. Therefore, the main objective of the research is to conduct an evaluation of the detection accuracy of pulp and pulp stones using a U2-Net-based CNN model.

## 2. Materials and Methods

All research was conducted in adherence to the Helsinki Declaration of Human Rights guidelines and ethical approval was obtained from the Cyprus International University Ethics Committee (Approval Number EKK23-24/005/08). The study aimed to achieve the segmentation of pulps and pulp stones by employing a U2-Net-based deep learning approach. A random process was used to divide the dataset into three sets, with 10% allocated to the validation set, 5% to the testing set, and the other 85% to train the model.

Panoramic radiographs were searched retrospectively from the clinic archives. Exclusion criteria included radiographs identified as having reduced quality, the presence of artifacts, or the involvement of deciduous teeth. As a result of this process, a total of 713 anonymized panoramic radiographs were deemed to be suitable for inclusion. A Newtom GO 3D/2D (Quantitive Radiology s.r.l., Verona, Italy) panoramic imaging device was used for taking all of the radiographs that were included in the study. The parameters used when taking the images were 80 kvP, 8 mA, with an exposure time of 14.2 s. The subsequent conversion of these images into PNG files enabled them to be later uploaded onto the Computer Vision Annotation Tool (CVAT v1.7.0 https://www.cvat.ai/) online tool for the examiners to apply labelling. All erupted permanent teeth on panoramic X-rays were included in the evaluation. Distinct, clearly defined radiopaque masses in the pulp chamber or root canal space were recorded as pulp stones. They were visible as separate, either rounded or irregular calcifications, distinctly detached from the canal walls and not connected to the dentinal structures; in this way, we differentiated the pulp stones from any root canal narrowing caused by aging, caries, or occlusal forces. Two examiners independently assessed the radiographs and labeled the pulps and pulp stones on the images using CVAT v1.7.0. The third examiner was also available to settle any disputes. Each observer reassessed 50 radiographs that were randomly selected after a month. Inter-examiner and intra-examiner agreement scores were evaluated using the Kappa score.

This study involved the development of two distinct machine learning models targeting the segmentation of different dental structures in 2D panoramic radiographs. The models were designed to segment dental pulp and pulp stone, respectively. Both models used the U2-Net (U square net) architecture, a deep learning architecture known for its efficacy in semantic segmentation tasks, introduced in Pattern Recognition, 2020. Various other architectures were also tested before settling on the U2-Net architecture, such as U-Net and Res-UNet. Within the same baseline parameters, the U2-Net architecture performed best, so it was picked for fine-tuning.

### 2.1. Model Pipeline

The workflow used in this research includes the following steps: First, the medical images were preprocessed. Then, each pixel within the images was classified into pulp and pulp stone using their respective models. Finally, the pulp and pulp stone segmentation maps were extracted for further analysis.

### 2.2. Dataset and Preprocessing

The datasets comprised 2D panoramic radiographs, with 462 images utilized for Pulp Stone Segmentation and 220 images for Pulp Segmentation. Normalization was performed using a simple min–max normalization within a fixed window. An examination of other techniques was also conducted. Based on the experimental results, the preprocessing method selected does not impact the training process and similar outcomes are largely achieved regardless of the approach taken. The division of the dataset was performed on a random basis, with 5%, 10%, and 85% allocated for validation, testing, and training, respectively. For augmentation, horizontal flipping was applied with a probability of 0.5 to enrich the dataset and increase robustness.

### 2.3. Semantic Segmentation

Semantic segmentation involves the assignment of a classification label to the individual pixels within an image. For the purpose of classifying pixels as belonging to either pulp, pulp stone or background, the U2-Net architecture was used, which is an extension of the U-Net model. U2-Net uses what is called an encoder−decoder architecture. Semantic information is captured by the model via the encoding path (or down convolution), whereas spatial information is recovered via the decoding path (upsampling). In this way, high-level semantic information can be effectively captured by U2-Net, while also ensuring that all spatial details required to accurately classify pixels are still preserved (Figure 1 and Figure 2). One of the issues in panoramic radiography segmentation is the difficulty of capturing small, low-contrast objects such as pulp stones among complex surrounding structures. U-Net has proven effective for many segmentation tasks; however, its single-scale expansion and contraction path can struggle with fine details. U2-Net addresses this by stacking multiple U-shaped modules at different scales, providing even deeper supervision and more nuanced feature extraction without drastically increasing the model size. This design is particularly suitable for panoramic images, where subtle differences in grayscale intensity can greatly affect segmentation accuracy.

### 2.4. Model Configuration

Both models use U2-Net architecture, with an input shape of 512 × 1024 × 1. The models used deep supervision techniques to enhance learning details at multiple scales in the architecture. The batch size was configured to 4 to optimize GPU memory usage. In order to prevent overfitting, the Adam optimizer was used with a learning rate and weight of decay of 0.0002 and 0.0001, respectively. As for the loss function, the dice coefficient was used. Python/JAX implementation of U2-Net was used as the basis for the algorithm in the current study. An NVIDIA^®^ GeForce^®^ RTX 3090 GPU was used when conducting the training and experimental stages. For the Pulp Stone Segmentation task, the model was trained for 500 epochs, whereas the Pulp Segmentation model was trained for 250 epochs.

## 3. Results

Before discussing our quantitative results, we will define the evaluation metrics utilized in this study. Intersection over Union (IoU) is a measurement of the spatial overlap between the predicted and ground truth segmentations, calculated as the ratio of their intersection area to their union area, with values ranging from 0 to 1, with 1 being a perfect match. The Dice coefficient, equivalent to the F1-score in binary classification contexts, quantifies segmentation accuracy as 2|A∩B|/(|A| + |B|), where A represents the predicted set and B is the ground truth set. This results in values ranging from 0 to 1, with 1 being a perfect match. Precision reflects the model’s positive predictive value, while recall characterizes its sensitivity in detecting the target structures.

Upon evaluation with the test dataset, the Dice coefficient and Intersection over Union (IoU) values recorded for the Pulp Segmentation model were 0.84 and of 0.758, respectively. Precision and recall were computed to be 0.858 and 0.827, respectively. The Pulp Stone Segmentation model achieved a Dice coefficient of 0.759 and an IoU of 0.686, with precision and recall at 0.792 and 0.773, respectively. These metrics indicate the models’ capabilities at accurately segmenting the desired dental structures (Figure 3).

## 4. Discussion

This study developed a highly effective artificial intelligence algorithm based on the U2-Net architecture to automatically segment pulp and pulp stones in dental panoramic images, as we hypothesized. AI is increasingly being recognized as a transformative force in dentistry, offering unprecedented opportunities for enhancing diagnostic accuracy and treatment planning. In recent years, AI models—especially those leveraging deep learning—have successfully analyzed medical images across various fields, including dermatology, radiology, and ophthalmology. AI is still in a relatively early phase in dentistry, but has shown promising results in detecting caries, periodontal disease, restorations, and periapical lesions. As artificial intelligence has become more advanced, the ability to analyze dental and medical images has been significantly enhanced via the usage of U2-Net and deep learning algorithms [20,21,22,23,24,32,33,34,35,36]. Among these applications, the automated detection of pulp stones remains underexplored despite its clinical importance, particularly in routine dental imaging such as panoramic radiography.

Intraoral radiographs, including periapical and bitewing images, provide a significantly higher spatial resolution than panoramic radiographs. This enhanced resolution makes them more appropriate for identifying fine details such as pulp stones, interproximal caries, and early-stage periodontal bone loss. However, their limited field of view restricts them to localized diagnostics [37,38].

On the other hand, panoramic radiography offers a comprehensive view of the entire maxillofacial region, including the jaws, dentition, temporomandibular joints, and surrounding anatomical structures. Although the resolution is lower and small calcifications may be overlooked due to overlapping anatomical features, panoramic X-rays are extremely valuable for initial assessments, treatment planning, and epidemiological screenings in public health settings because of their speed, reduced radiation dose, and patient comfort [39,40].

Recent studies indicate that combining panoramic radiographs with AI-based post-processing or additional intraoral imaging significantly improves diagnostic sensitivity and specificity, especially in detecting calcified structures such as pulp stones or carotid artery calcifications [41,42].

Thus, each imaging modality has its strengths: intraoral radiographs excel in detail and precision for specific regions. In contrast, panoramic radiographs offer a global overview crucial for screening and comprehensive dental evaluations. Clinical objectives, patient-specific needs, and diagnostic requirements should guide the choice between them.

This research focuses on resolved an important deficiency in the existing literature by introducing an AI-based segmentation model tailored to identify pulp and pulp stones in panoramic radiographs. While previous studies have utilized deep learning approaches to identify pulp stones on bitewing images or segment pulp in CBCT scans [11,14,43,44,45], according to our understanding, the current study represents the first attempt to conduct an evaluation of the segmentation of pulp and pulp stones utilizing the U2-Net architecture in 2D panoramic imaging. This is particularly important as panoramic radiographs are far more accessible in routine dental settings for assessing the full dentition, providing a broader anatomical overview with greater patient comfort. It is preferred for screening, especially in large-scale dental practices and public health institutions.

From a technical standpoint, our study demonstrates that the U2-Net architecture is well-suited for the segmentation of fine structures like pulp stones at the pixel level. The network’s multi-scale residual blocks and deep supervision allow it to capture global context and fine-grained details—essential features for distinguishing small calcifications within complex anatomical backgrounds. The achieved Dice coefficient of 0.84 for pulp and 0.759 for pulp stones, along with IoU scores of 0.758 and 0.686, respectively, suggest that the model performs well despite the inherent challenges of the input data. In particular, pulp stones—being small, irregular, and sometimes poorly contrasted—represent a challenging target, and the ability of our AI system to accurately delineate them is noteworthy.

While CBCT can accurately determine the prevalence of pulp stones, its routine utilization for detecting pulp stones should be discouraged due to its higher radiation exposure compared to two-dimensional radiography [46].

Comparing our findings with prior works highlights our approach’s promise and uniqueness. For instance, Yuce et al. [11] used YOLOv4 to detect pulp calcifications in bitewing radiographs, achieving a precision of 98.94%. Altındağ et al. [14] used a Mask R-CNN model and achieved an F1-score of 0.8995. While our model’s scores are slightly lower, this should be interpreted within the context of the imaging modality—panoramic images contain significantly more anatomical noise, lower contrast, and wider variation. CBCT-based models, such as those by Duan et al. [44] and Lin et al. [45], offer higher precision due to the 3D nature and higher resolution of the input data, but their applicability in routine settings is limited by radiation dose, cost, and availability.

On the other hand, Selmi et al. used a CNN-based feature extraction approach along with multiple classifiers for the purpose of detecting pulp stones in periapical radiographs [47]. According to their findings, when the Medium Gaussian Support Vector Machine was used, the accurate rate achieved by the Residual Network 50 was 76.4%, while the accuracy was 73.1% for Inception v3 with the identical classifier.

These findings differ from our findings, as they found higher percentages. This might be attributed to the fact that they used more radiographs than our study.

A study by Boztuna et al. [48] demonstrated that a U2-Net-based AI model can effectively detect periapical lesions on panoramic radiographs, achieving strong performance metrics on the validation set and precision, recall, and F1-score of 0.82, 0.77, and 0.8, respectively, on the test set. These results align with ours using the U2-Net-based AI model.

Beyond its technical merits, this study contributes to the broader vision of incorporating AI into daily clinical practice. Panoramic radiographs are ubiquitous in general dentistry, and integrating AI into existing imaging software could provide real-time decision support, highlight suspicious regions, and offer quantifiable insights for endodontic planning. This has immense value in both private clinics and teaching hospitals. Furthermore, such systems can reduce the cognitive burden on clinicians, improve diagnostic standardization, and serve as second-opinion tools in ambiguous cases.

Another important application of this technology lies in dental education. Novice dental students and early-career practitioners often lack the experience to detect subtle radiographic findings such as pulp stones. Educational institutions can integrate AI into training platforms to enhance radiographic interpretation skills. Real-time feedback and visual AI overlays may boost learning efficiency and diagnostic accuracy, supporting modern, technology-driven dental education.

Despite these promising outcomes, the study has certain limitations. The radiographic dataset, although extensive, was collected from a single institution using a specific imaging system, which may limit generalizability. Including negative cases (images without pulp stones) introduces class imbalance, which may influence model performance when deployed in a broader clinical context. Additionally, while the U2-Net model performed well in our tests, further validation across multi-center datasets with varied imaging hardware is necessary before clinical deployment. Another limitation is that the model focused solely on segmentation tasks; future studies could combine segmentation with classification layers to provide comprehensive diagnostic reports or severity assessments. Furthermore, the diagnostic performance of the AI model was evaluated using expert-labeled panoramic radiographs as the reference standard, rather than a gold standard modality such as cone-beam computed tomography (CBCT) or histological validation.

Looking ahead, there are several directions for further research. First, future work should assemble larger, more diverse datasets, including multiple imaging systems and patient demographics, to improve model generalizability. Second, integrating the model into a clinical decision support system (CDSS) that offers endodontic treatment planning suggestions, alerts for calcification-related complications, and historical comparisons could be transformative. Third, including uncertainty quantification in AI outputs would help clinicians assess confidence levels in predictions, an essential factor in real-world decision-making. Finally, newer architectures, including hybrid transformer−CNN models, should be explored for further performance enhancement. To improve future diagnostic accuracy, combining panoramic images with clinical data or intraoral imaging could provide a more detailed diagnostic framework. Incorporating CBCT or histological confirmation in some instances would allow for a more accurate evaluation of the model performance as a gold standard.

This study demonstrates the successful development of a deep learning model capable of segmenting pulp and pulp stones on panoramic radiographs—a first in dental AI research using U2-Net. The findings highlight not only the technical feasibility of such models, but also their clinical relevance and educational value. With the continued evolution of AI, if it is carefully integrated into daily dental diagnostic practices, it can potentially revolutionize patient care, improve outcomes, and elevate the standard of dental education worldwide.

## 5. Conclusions

This investigation developed a highly effective artificial intelligence algorithm for automatically segmenting pulp and pulp stones in panoramic radiographic images using the U2-Net architecture. The model demonstrates promise for inclusion in clinical workflows, providing fast and accurate measurements essential for diagnostic and therapeutic applications.

## Figures and Tables

**Figure 1 dentistry-13-00274-f001:**
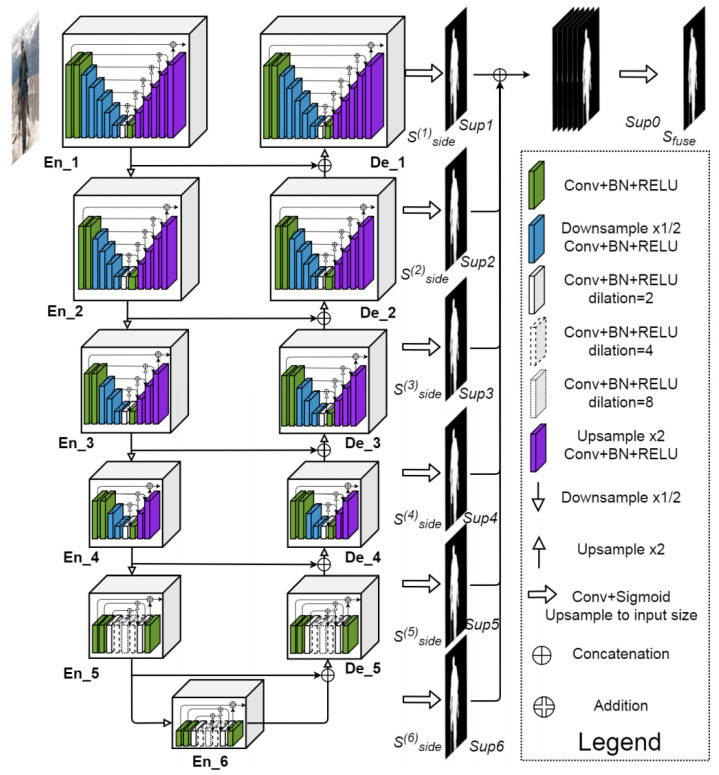
U2-Net architecture.

**Figure 2 dentistry-13-00274-f002:**
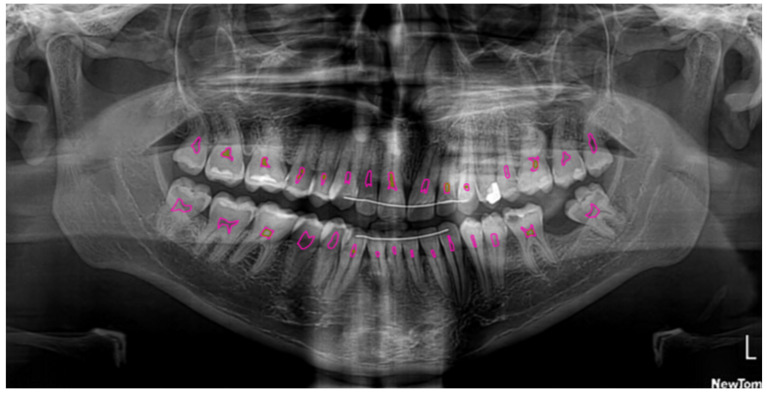
Segmentation of the pulp (pink) and pulp stones (green) using U2-Net.

**Figure 3 dentistry-13-00274-f003:**
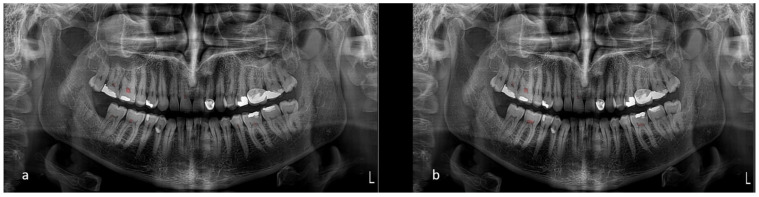
(**a**) Model’s segmentation and (**b**) ground truth.

## Data Availability

The datasets used and/or analyzed during the current study are available from the corresponding author upon reasonable request.

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
