# Peer review of "Segmentation of Pulp and Pulp Stones with Automatic Deep Learning in Panoramic Radiographs: An Artificial Intelligence Study"

_dentistry, 2025, doi:10.3390/dj13060274_

Round 1

Reviewer 1 Report

Comments and Suggestions for Authors

I commend the authors for their diligent and forward-thinking work in exploring the diagnostic potential of deep learning for segmenting pulp and pulp stones in panoramic radiographs. This manuscript represents a timely and meaningful contribution to the field of dental radiology and artificial intelligence, particularly given the clinical significance of pulp stone detection and the technical challenges posed by panoramic imaging.

The study is grounded in a clear and well-defined objective, and the authors have employed a robust methodology using the U²-Net architecture. The segmentation performance: quantified through Dice, IoU, precision, and recall metrics, is convincingly strong, especially given the inherent noise and complexity of panoramic datasets. The thoughtful use of multiple evaluators for ground truth annotation, along with inter- and intra-examiner validation, adds a layer of rigor and reproducibility that is often underemphasized in similar studies.

The discussion demonstrates strong scientific reasoning, especially in the candid comparison with previous models and imaging modalities. The limitations are acknowledged transparently, and the future directions suggested reflect a forward-looking mindset that prioritizes both technical refinement and clinical integration.

That said, I believe the introduction, while informative, can benefit from more focused elaboration on the gap in current AI applications in panoramic radiography and more contextualization of the U²-Net’s uniqueness among segmentation models. Strengthening this part would better highlight the novelty of the study.

Overall, this work is technically sound, clinically relevant, and written with clarity. It opens promising avenues for AI-assisted diagnostic tools in dental practice and education.

Author Response

  1. I believe the introduction, while informative, can benefit from more focused elaboration on the gap in current AI applications in panoramic radiography and more contextualization of the U²-Net’s uniqueness among segmentation models. Strengthening this part would better highlight the novelty of the study.
    We agree with the reviewer, and we added a paragraph to the introduction section as; “AI has been applied to improve the analysis of dental radiographic images. In the context of two-dimensional (2D) radiographs, these digital images consist of numerous pixels, each with different levels of brightness and radiopacity. AI systems are trained to interpret these images by learning from these pixel-based features [26-27]. Recent advances in AI have shown promise in medical imaging; a deep learning algorithm capable of automatically detecting teeth in panoramic radiographs is regarded as a significant advancement in dental practice [28-29]. Also, lesion diagnosis using AI and automatic description of anatomic landmarks showed great results in terms of AI and panoramic radiographs [26,30,31].” Also we added statements to the materials and methods section as “One of the issues in panoramic radiography segmentation is the difficulty of capturing small, low-contrast objects such as pulp stones among complex surrounding structures. U-Net has proven effective for many segmentation tasks however its single-scale expansion and contraction path can struggle with fine details. The U2-Net addresses this by stacking multiple U-shaped modules at different scales, providing even deeper supervision and more nuanced feature extraction without drastically increasing model size. This design is particularly suitable for panoramic images, where subtle differences in grayscale intensity can greatly affect segmentation accuracy. “

Reviewer 2 Report

Comments and Suggestions for Authors

The paper by Mujgan Firincioglulari et al. entitled “Segmentation of Pulp and Pulp Stones with Automatic Deep Learning in Panoramic Radiographs: An Artificial Intelligence Study” is an interesting study. However, I have the following concerns.

  1. Typically, the presence or absence of pulp stones is not a primary treatment target. Dental X-rays are preferred over panoramic X-rays for detecting pulp stones during root canal treatments due to higher diagnostic accuracy. Please justify the validity of using panoramic X-rays combined with AI for pulp stone detection.
  2. Without a gold standard reference (such as CBCT or microscopic observation), it is not feasible to accurately quantify the diagnostic accuracy of the AI method used in this study.

  1. Please clarify the purpose of using panoramic X-ray photographs in the study (Implants, orthodontics, temporomandibular joint disorders, etc.)

  1. Did you perform a sample size analysis? If so, please provide details.

  1. Specify the average age, gender ratio, and ethnicity of the patient population.

  1. Clearly define the criteria used to identify pulp stones on panoramic X-ray images.

  1. Distinguish pulp stones explicitly from root canal narrowing caused by caries, occlusion, or aging.

  1. In Figure 2, the left mandibular first molar exhibits caries, suggesting a different condition from healthy teeth. Clearly state your inclusion and exclusion criteria for tooth selection.

  1. The superscripts of “2” associated with U2-Net are inconsistent; please standardize them.

  1. References 2 and 3 are identical; please correct this duplication.

  1. Reference numbers are duplicated (e.g., "1. 1. 2. 2."). Please correct these errors.

  1. Include your speculation on why the current AI model demonstrated superior performance and discuss potential improvements for future diagnostic accuracy.

  1. Provide a comparison of diagnostic accuracy between panoramic X-rays and dental imaging in more detail.

Author Response

  1. Typically, the presence or absence of pulp stones is not a primary treatment target. Dental X-rays are preferred over panoramic X-rays for detecting pulp stones during root canal treatments due to higher diagnostic accuracy. Please justify the validity of using panoramic X-rays combined with AI for pulp stone detection.
    We thank the reviewer for their valuable observation about the traditional diagnostic method for pulp stones. We agree that dental (intraoral) radiographs are preferred over panoramic radiographs due to their better resolution and diagnostic accuracy in endodontic treatment contexts. As mentioned in the manuscript, panoramic radiographs are far more accessible in routine dental settings and are preferred for screening, especially in large-scale dental practices and public health institutions. Although pulp stones are generally not primary targets in endodontic treatment, identifying them can offer considerable epidemiological and diagnostic insights.
  2. Without a gold standard reference (such as CBCT or microscopic observation), it is not feasible to accurately quantify the diagnostic accuracy of the AI method used in this study.

We recognize that the absence of a definitive gold standard limits our ability to quantify the diagnostic accuracy of the AI model. In our research, we compared the AI predictions to expert-labeled panoramic radiographs. Our aim was not to position AI as a substitute for gold-standard methods like CBCT or histological evaluation. Instead, we sought to investigate the practicality of AI-assisted detection of pulp stones in commonly obtained panoramic radiographs, especially for large-scale epidemiological studies or screening.  In this regard, we added this as a limitation to the discussion section, as” the diagnostic performance of the AI model was evaluated using expert-labeled panoramic radiographs as the reference standard, rather than a gold standard modality such as cone-beam computed tomography (CBCT) or histological validation.”

  1. Please clarify the purpose of using panoramic X-ray photographs in the study (Implants, orthodontics, temporomandibular joint disorders, etc.)

We appreciate the reviewer’s request for clarification regarding the purpose of using panoramic X-ray photographs in our study.

The main reason for using panoramic radiographs was to assess the full dentition, including impacted teeth, during dental examinations.

  1. Did you perform a sample size analysis? If so, please provide details.

We did not conduct a formal sample size analysis. Since this was a retrospective analysis of routinely acquired panoramic radiographs, our primary goal was to make the most of the available dataset to enhance statistical power and improve model training efficiency. The dataset size was established based on the number of eligible panoramic images gathered during the specified period.  

  1. Specify the average age, gender ratio, and ethnicity of the patient population.

Since the primary objective of this study was to evaluate the feasibility of detecting pulp stones using artificial intelligence applied to panoramic radiographs, demographic information was not collected, as it was not expected to influence the image-based detection process.

  1. Clearly define the criteria used to identify pulp stones on panoramic X-ray images.

Distinct, clearly defined radiopaque masses in the pulp chamber or root canal sapce are recorded as pulp stones and we added this statement to the materials and methods.

  1. Distinguish pulp stones explicitly from root canal narrowing caused by caries, occlusion, or aging.

Pulp stones were identified as distinct, clearly defined radiopaque masses within the pulp chamber or root canal space. They were visible as separate, either rounded or irregular calcifications, distinctly detached from the canal walls and not connected to the dentinal structures.

Conversely, root canal narrowing caused by aging, caries, or occlusal forces generally manifests as a gradual, widespread decrease in canal width, frequently impacting the entire length of the canal and demonstrating continuity with adjacent dentin. Unlike pulp stones, these alterations do not exhibit distinct boundaries.

We added this statement: "Distinct, clearly defined radiopaque masses in the pulp chamber or root canal space are recorded as pulp stones. They were visible as separate, either rounded or irregular calcifications, distinctly detached from the canal walls and not connected to the dentinal structures; in this way, we differentiate the pulp stones from root canal narrowing caused by aging, caries, or occlusal forces,” to the materials and methods section.

  1. In Figure 2, the left mandibular first molar exhibits caries, suggesting a different condition from healthy teeth. Clearly state your inclusion and exclusion criteria for tooth selection.

Since the primary objective of this study was to evaluate the feasibility of detecting pulp stones using artificial intelligence applied to panoramic radiographs, we did not distinguish between carious and non-carious teeth. We included and assessed all erupted permanent teeth in the panoramic X-rays to detect pulp stones and added a statement about this to the materials and methods section, as “All erupted permanent teeth on panoramic X-rays are included in the evaluation. “

  1. The superscripts of “2” associated with U2-Net are inconsistent; please standardize them.

We standardize them.

  1. References 2 and 3 are identical; please correct this duplication.

We removed reference 2 and revised the references.

  1. Reference numbers are duplicated (e.g., "1. 1. 2. 2."). Please correct these errors.

We corrected the errors

  1. Include your speculation on why the current AI model demonstrated superior performance and discuss potential improvements for future diagnostic accuracy.

We added sentences as” U-Net has proven effective for many segmentation tasks however its single-scale expansion and contraction path can struggle with fine details. The U2-Net addresses this by stacking multiple U-shaped modules at different scales, providing even deeper supervision and more nuanced feature extraction without drastically increasing model size. This design is particularly suitable for panoramic images, where subtle differences in grayscale intensity can greatly affect segmentation accuracy. “ to the materials section and “To improve future diagnostic accuracy, combining panoramic images with clinical data or intraoral imaging could provide a more detailed diagnostic framework. Incorporating CBCT or histological confirmation in some instances would allow for a more accurate evaluation of model performance as a gold standard.  “ to the discussion section for future diagnostic accuracy

  1. Provide a comparison of diagnostic accuracy between panoramic X-rays and dental imaging in more detail.

We added a comparison to the discussion section as “ Intraoral radiographs, including periapical and bitewing images, provide significantly higher spatial resolution than panoramic radiographs. This enhanced resolution makes them more appropriate for identifying fine details such as pulp stones, inter-proximal caries, and early-stage periodontal bone loss. However, their limited field of view restricts them to localized diagnostics [37][38].

On the other hand, panoramic radiography offers a comprehensive view of the entire maxillofacial region, including the jaws, dentition, temporomandibular joints, and sur-rounding anatomical structures. Although the resolution is lower and small calcifications may be overlooked due to overlapping anatomical features, panoramic X-rays are extremely valuable for initial assessments, treatment planning, and epidemiological screenings in public health settings because of their speed, reduced radiation dose, and patient comfort [39-40].

Recent studies indicate that combining panoramic radiographs with AI-based post-processing or additional intraoral imaging significantly improves diagnostic sensi-tivity and specificity, especially in detecting calcified structures such as pulp stones or carotid artery calcifications [41-42].

Thus, each imaging modality has its strengths: intraoral radiographs excel in detail and precision for specific regions. In contrast, panoramic radiographs offer a global overview crucial for screening and comprehensive dental evaluations. Clinical objectives, patient-specific needs, and diagnostic requirements should guide the choice between them.”

Reviewer 3 Report

Comments and Suggestions for Authors

I am happy to evaluate your hard work, 

here some comments to improve your study.

Keywords: Pulp stone; artificial intelligence; panoramic radiograph; deep learning

place them in alphabetical order

Introduction:

line 60: Artificial intelligence [AI] can help to minimize the misdiagnosis of pulp stones.

the transition was not smooth, you can add this sentence after you introduce the AI as in the following paragraph

line 81: using a CNN model based on the U2-Net architecture

would you describe the 2 systems in the introduction

check: https://www.hellopearl.com/products/second-opinion  , 

what is your hypothesis?

Material and methods:

Line 88: The dataset was randomly divided, 10% was used for validation, 5% for testing, 88 and the remaining 85% for training.

where did you get this formula? explain.

Line 93:

All included radiographs were taken using the Newtom GO Pulps and pulp stones on these radiographs were 3D/2D [Quantitive Radiology s.r.l., Verona, Italy] panoramic imaging system.

revise this sentence> 3D/2D???

LINE96: Images were obtained in DICOM format

IS IT CBCT?  DICOM format is not for panoramic? explain or correct.

RESUTS:

Not easy to understand, try to make it simpler.

Discussion:

at the begining 1st or second paragraph you should mention what did you found and if its what you expected .... as hypothized?

Pulp stones are small calcified structures that develop within the dental pulp. Alt-183 hough they are often asymptomatic, their presence can complicate root canal therapy by 184 obstructing canal orifices, deflecting instruments, or contributing to treatment failure if 185 overlooked. Their detection is particularly relevant to endodontists and general dentists 186 performing endodontic procedures. Accurately identifying these structures can also aid 187 in risk assessment for dental treatments and in understanding potential associations with 188 systemic conditions such as cardiovascular disease [7–12,33].  THIS PART ALMOST A REPITION FROM THE INTRODUCTION. REMOVE IT OR JUST CLARIFY YOUR MESSAGE

line 222: On the other hand, Selmi et al. [2022] detect pulp stones in dental radiographs using

WHAT RADIOGRAPH?

line 243-360 : too much and its a bit away from your study, just make it short

ANY ARTICLES USED THE SAME MODELS ON ANY RADIOGRAPH, SHOULD BE ADDED HERE TO COMPARE THE ACCURACY OF YOUR MODEL YOU USED.

CONCLUSION:

This investigation developed a highly effective artificial intelligence algorithm for automatically segmenting pulp and pulp stones in dental radiographic images using the  U2-Net architecture.  ADD "PANORAMIC" radiographic image.

References: 

2 and 3 are the same

Author Response

  1. Keywords: Pulp stone; artificial intelligence; panoramic radiograph; deep learning, place them in alphabetical order

Corrected.

  1. Introduction:

line 60: Artificial intelligence [AI] can help to minimize the misdiagnosis of pulp stones.

the transition was not smooth, you can add this sentence after you introduce the AI as in the following paragraph

We agree with the reviewer and have made the suggested correction.

  1. line 81: using a CNN model based on the U2-Net architecture

would you describe the 2 systems in the introduction

check: https://www.hellopearl.com/products/second-opinion,

There is only one system described, which is based on a CNN model inspired by the U²-Net architecture.

  1. what is your hypothesis?

We added a sentence about hypothesis to the introduction section as “We hypothesize that artificial intelligence can precisely identify pulp stones in routine panoramic radiographs, achieving diagnostic performance similar to that of expert human interpretation, despite the inherent limitations of panoramic imaging.”

  1. Material and methods:

Line 88: The dataset was randomly divided, 10% was used for validation, 5% for testing, 88 and the remaining 85% for training. where did you get this formula? explain.

We revised the sentence as “A random process was used to divide the dataset into three sets, with 10% allocated to the validation set, 5% to the testing set, and the other 85% was used to train the model. “ This formula is reasonable and often used, but not a universal rule or formula". It's just one possible configuration based on experimentation or preference.

  1. Line 93:

All included radiographs were taken using the Newtom GO Pulps and pulp stones on these radiographs were 3D/2D [Quantitive Radiology s.r.l., Verona, Italy] panoramic imaging system.

 revise this sentence> 3D/2D???

We revised the sentence as “ All included radiographs were taken using the Newtom GO 3D/2D [Quantitive Radiology s.r.l., Verona, Italy] panoramic imaging system. “

  1. LINE96: Images were obtained in DICOM format

IS IT CBCT?  DICOM format is not for panoramic? explain or correct.

We corrected the sentence as “ Images were converted into PNG files.”

  1. RESUTS:

We tried to explain it in the simplest way we could

  1. Discussion:

at the begining 1st or second paragraph you should mention what did you found and if its what you expected .... as hypothized?

We agree with the reviewer on the issue and mentioned what we found in the first paragraph of the discussion: "This study developed a highly effective artificial intelligence algorithm based on the U²-Net architecture to segment pulp and pulp stones in dental panoramic images automatically, as we hypothesized. “

  1. Pulp stones are small calcified structures that develop within the dental pulp. Alt-183 hough they are often asymptomatic, their presence can complicate root canal therapy by 184 obstructing canal orifices, deflecting instruments, or contributing to treatment failure if 185 overlooked. Their detection is particularly relevant to endodontists and general dentists 186 performing endodontic procedures. Accurately identifying these structures can also aid 187 in risk assessment for dental treatments and in understanding potential associations with 188 systemic conditions such as cardiovascular disease [7–12,33]. THIS PART ALMOST A REPITION FROM THE INTRODUCTION. REMOVE IT OR JUST CLARIFY YOUR MESSAGE

We agree with the reviewer on the issue and removed the paragraph.

  1. line 222: On the other hand, Selmi et al. [2022] detect pulp stones in dental radiographs using

WHAT RADIOGRAPH?

We corrected as periapical radiographs.

  1. line 243-360 : too much and its a bit away from your study, just make it short

We shortened as “Educational institutions can integrate AI into training platforms to enhance radiographic interpretation skills. Real-time feedback and visual AI overlays may boost learning efficiency and diagnostic accuracy, supporting modern, technology-driven dental education.”

  1. ANY ARTICLES THAT USE THE SAME MODELS ON ANY RADIOGRAPH SHOULD BE ADDED HERE TO COMPARE THE ACCURACY OF THE MODEL YOU USED.

We added a paragraph to the discussion section as ”A study by Boztuna et al. [48] demonstrates that a U2-Net-based AI model can effectively detect periapical lesions on panoramic radiographs, achieving strong performance metrics on the validation set and precision, recall, and F1-score of 0.82, 0.77, and 0.8, respectively, on the test set. These results align with ours using the U2-Net-based AI model.”

  1. CONCLUSION:

This investigation developed a highly effective artificial intelligence algorithm for automatically segmenting pulp and pulp stones in dental radiographic images using the  U2-Net architecture.  ADD "PANORAMIC" radiographic image.

We agree with the reviewer on the issue and made the correction.

  1. References:

2 and 3 are the same

Corrected

Round 2

Reviewer 2 Report

Comments and Suggestions for Authors

Thank you for your revisions. 
They are appropriate as provided, and no further changes are required.

Reviewer 3 Report

Comments and Suggestions for Authors

Thank you for the good work